# Replication-Deficient Lymphocytic Choriomeningitis Virus-Vectored Vaccine Candidate for the Induction of T Cell Immunity against *Mycobacterium tuberculosis*

**DOI:** 10.3390/ijms23052700

**Published:** 2022-02-28

**Authors:** Elodie Belnoue, Alexis Vogelzang, Natalie E. Nieuwenhuizen, Magdalena A. Krzyzaniak, Stephanie Darbre, Mario Kreutzfeldt, Ingrid Wagner, Doron Merkler, Paul-Henri Lambert, Stefan H. E. Kaufmann, Claire-Anne Siegrist, Daniel D. Pinschewer

**Affiliations:** 1Department of Pathology and Immunology, University of Geneva, 1211 Geneva 4, Switzerland; elodie.belnoue@gmail.com (E.B.); stephanie.darbre@gmail.com (S.D.); mario.kreutzfeldt@unige.ch (M.K.); ingrid.wagner@unige.ch (I.W.); doron.merkler@unige.ch (D.M.); paul.lambert@unige.ch (P.-H.L.); claire-anne.siegrist@unige.ch (C.-A.S.); 2W.H.O. Collaborating Centre for Vaccine Immunology, University of Geneva, 1211 Geneva 4, Switzerland; 3Department of Immunology, Max Planck Institute for Infection Biology, 10117 Berlin, Germany; arvogelzang@gmail.com (A.V.); natalie.nieuwenhuizen@uni-wuerzburg.de (N.E.N.); kaufmann@mpiib-berlin.mpg.de (S.H.E.K.); 4Division of Experimental Virology, Department of Biomedicine, University of Basel, 4003 Basel, Switzerland; mkrzyzan@outlook.com; 5Division of Clinical Pathology, Geneva University Hospital, 1211 Geneva 4, Switzerland

**Keywords:** *Mycobacterium tuberculosis*, vaccine, lymphocytic choriomeningitis virus (LCMV), T cell immunity, neonatal vaccination, replication-deficient LCMV vector, Bacille Calmette Guerin (BCG) heterologous prime-boost

## Abstract

*Mycobacterium tuberculosis* (*Mtb*) represents a major burden to global health, and refined vaccines are needed. Replication-deficient lymphocytic choriomeningitis virus (rLCMV)-based vaccine vectors against cytomegalovirus have proven safe for human use and elicited robust T cell responses in a large proportion of vaccine recipients. Here, we developed an rLCMV vaccine expressing the *Mtb* antigens TB10.4 and Ag85B. In mice, rLCMV elicited high frequencies of polyfunctional *Mtb*-specific CD8 and CD4 T cell responses. CD8 but not CD4 T cells were efficiently boosted upon vector re-vaccination. High-frequency responses were also observed in neonatally vaccinated mice, and co-administration of rLCMV with Expanded Program of Immunization (EPI) vaccines did not result in substantial reciprocal interference. Importantly, rLCMV immunization significantly reduced the lung *Mtb* burden upon aerosol challenge, resulting in improved lung ventilation. Protection was associated with increased CD8 T cell recruitment but reduced CD4 T cell infiltration upon *Mtb* challenge. When combining rLCMV with BCG vaccination in a heterologous prime-boost regimen, responses to the rLCMV-encoded *Mtb* antigens were further augmented, but protection was not significantly different from rLCMV or BCG vaccination alone. This work suggests that rLCMV may show utility for neonatal and/or adult vaccination efforts against pulmonary tuberculosis.

## 1. Introduction

Tuberculosis (TB) is a chronic infectious disease caused by the intracellular bacterial pathogen *Mycobacterium tuberculosis* (*Mtb*). Globally, an estimated 1.7 billion individuals are latently infected with *Mtb*, although infection can persist lifelong. These individuals are healthy and not contagious. In about 10% of these latently infected individuals, infection progresses to active disease. Whilst most disease cases emerge within the first 24 months, progression can also be delayed and precipitated by stress, aging or immune suppression. In 2020, 10 million individuals newly developed active TB disease and 1.5 million died [1]. The COVID-19 crisis has further worsened the situation, and it has been estimated that lower TB notification rates, lower drug treatment rates, and other obstacles caused by COVID-19 will lead to an additional 6.3 million cases of active TB disease and 1.4 million deaths by 2025 (WHO, 2021). TB can be treated and cured by chemotherapy; however, the treatment outcome is increasingly affected by multi-drug-resistant strains. The only currently available vaccine is a live-attenuated strain of *Mycobacterium bovis*, the agent of cattle TB, termed Bacille Calmette Guérin (BCG) [2]. It is generally accepted that BCG prevents extrapulmonary and disseminated TB in infants but shows insufficient protection against pulmonary TB, the major form of disease in all age groups [3]. Since BCG is a viable vaccine, immunosuppression can lead to disseminated disease called BCG-osis. For these reasons, BCG is currently only recommended for infants who are HIV-negative. Therefore, general agreement exists that novel vaccines, which are safer and more efficient in all age groups, are urgently needed, notably to fulfill the target proposed by WHO for 2030 to reduce TB deaths and new cases of active TB by 95% or 90%, respectively.

The technological advances in molecular biology and recombinant vaccine technology of the 21st century have sparked renewed interest and optimism in the scientific community for better TB vaccines to be generated [4]. Accordingly, strong efforts addressing the various options are ongoing at the global scale. Amongst others, new candidates comprise adjuvanted protein subunit vaccines, live-attenuated; killed mycobacterial vaccines; and virally vectored delivery systems. A recent study on the adjuvanted protein vaccine candidate M72/AS01_E_ provided evidence for ca. 50% efficacy in preventing pulmonary tuberculosis in latently infected individuals [5]. Revaccination with BCG resulted in ca. 50% lower rates of sustained infection [6]. 

While a key role of CD4 T cells in tuberculous granuloma formation and immune control during latency has long been widely accepted, the advent of lymphocyte subset depletion and mouse knock-out models has attracted broader interest in CD8 T cells, and several observations in experimental mouse models suggest a contribution of these cells to *Mtb* control and disease prevention [7,8,9]. The corroboration of such findings in non-human primate (NHP) models [10] confirms CD8 T cells as an important correlate of protection to be targeted by vaccination strategies [11,12].

Lymphocytic choriomeningitis virus (LCMV) has long been known to elicit CD8 T cell responses of exceptional magnitude, functionality, and longevity, and generations of immunologists have therefore exploited the LCMV infection model to study T cell immunobiology [13,14,15,16]. With the advent of reverse genetic techniques for engineering of the infectious virus’ genome [17,18,19], it has become possible to convert LCMV into an apathogenic replication-deficient viral vaccine delivery platform, referred to herein as rLCMV [20]. rLCMV has proven highly immunogenic in mice, guinea pigs, and non-human primates (NHPs), inducing high-frequency CD8 T cell responses against various antigens [20,21,22,23]. In a recent first-in-human study, these findings have been extended to humans [24]. Not only did an rLCMV-based cytomegalovirus (CMV) vaccine candidate elicit CMV-specific antibody responses but also 93% of CMV-naïve healthy volunteers in the trial mounted substantial CD8 T cell responses to the vectorized CMV antigens. Another important differentiating feature of the rLCMV vector platform is the virtual absence of vector-neutralizing antibody (nAbs) induction even after multiple rounds of vaccination [20,23,24], which is due to the vector’s glycan-shielded envelope protein [25]. Accordingly, rLCMV can be re-administered repeatedly without eliciting substantial anti-vector antibody immunity, resulting in incremental increases in T cell responses against vectorized cargo [24]. These mechanistic observations were originally described in mice and have been confirmed in NHPs and humans [20,23,24]. This renders the rLCMV platform attractive for vaccination against infectious diseases, for which potent T cell immunity has to be induced and sustained by regular booster vaccination over the course of several years or even decades.

TB10.4 and Ag85B are two of the most immunodominant *Mtb* proteins, and they both have been included as antigens in multiple TB vaccine candidates tested in preclinical and clinical trials [26,27,28]. Ag85B is abundantly expressed by *Mtb* during the early stages of infection [29]. It has a critical role in mycobacterial cell wall synthesis and is also thought to play a role in immune evasion [27,30]. TB10.4 belongs to a subfamily of the early secretory antigenic target gene 6 (ESAT6) family and induces strong T cell responses in both animals and humans [26]. 

Here, we developed an rLCMV-based *Mtb* vaccine candidate delivering an artificial conjugate of the *Mtb* antigens Ag85B and TB10.4 (Ag85B-TB10.4) [31], characterized its immunogenicity in neonatal and adult mice, and tested its protective efficacy against an *Mtb* aerosol challenge. Altogether our findings position rLCMV as an attractive new vector for the delivery of *Mtb* antigens and as a candidate to be considered for inclusion in refined neonatal and adult *Mtb* vaccination regimens.

## 2. Results

### 2.1. rLCMV Vector Delivering Ag85B-TB10.4 Elicits High Frequencies of Polyfunctional CD8 and CD4 T Cells

We engineered a replication-deficient LCMV-based vaccine vector candidate (rLCMV, Figure 1A) by replacing the viral glycoprotein (GP) open reading frame with a fusion polypeptide consisting of the *Mtb* antigens Ag85B and TB10.4 [31] (Figure 1B). These antigens contain well-characterized CD4 and CD8 epitopes for the H-2^b^ haplotype of C57BL/6 mice, respectively. To assess the immunogenicity of the rLCMV, we administered the vector to C57BL/6 mice by either the subcutaneous (s.c.), intramuscular (i.m.) or intravenous (i.v.) route. We determined splenic CD8 and CD4 T cell responses 28 days later by MHC class I and class II multimer staining, respectively (Figure 1C,E), and by intracellular cytokine assays (Figure 1D,F). rLCMV i.v. immunization elicited TB10.4-specific CD8 T cell responses exceeding 10% of total CD8 T cells in spleen and Ag85B-specific responses in the range of 2–3% of CD4 T cells. All routes elicited detectable immune responses, but i.v. administration was significantly more immunogenic than the i.m. route, and we noted a trend for the latter to be superior to s.c. administration. Both CD8 and CD4 T cell responses exhibited polyfunctionality, with a proportion of the CD8 T cells co-producing IFN-γ and TNF-α, and some CD4 T cells secreting not only IFN-γ and TNF-α but also IL-2 upon peptide restimulation. 

### 2.2. rLCMV-Induced CD8 T Cell Responses Are Augmented by Homologous Boosting

Next, we studied whether rLCMV-induced CD8 and CD4 T cell responses could be augmented by homologous booster vaccination. These and most of the subsequent experiments relied on the i.m. administration route, which has been clinically validated for an rLCMV-based vaccine candidate against cytomegalovirus [24]. MHC class I multimer staining and intracellular cytokine assays following stimulation with Ag85B and TB10.4 peptides showed that a homologous boost delivered on day 28 significantly increased the frequency of *Mtb*-specific cells among CD8 T cell in spleen (Figure 2A). More than 20% of splenic CD8 T cells in prime-boost immunized mice responded to peptide stimulation, as opposed to ~3% in animals with prime alone. Moreover, the homologous boost increased the proportion of polyfunctional splenic CD8 T cells secreting IFN-γ and TNF-α in conjunction with surface expression of the lytic granule release marker CD107a (LAMP1, Figure 2B) to ~10% of CD8 T cells, thus markedly exceeding the ~2% of cells in mice having received only one dose of rLCMV. In contrast to this pronounced homologous booster effect on CD8 T cells, the analysis of Ag85B-specific CD4 T cell responses by either MHC class II tetramers or intracellular cytokine assays indicated at best a modest booster effect on CD4 T cell responses (Figure 2C,D). 

### 2.3. Neonatal Mice Mount Robust CD8 and CD4 T Cell Responses to rLCMV Vaccination

Prevention of tuberculosis in newborns represents a significant medical need, which prompted us to test the immunogenicity of rLCMV in 1-week-old mice, which can serve as a model for the immunological immaturity of human neonates [32]. We immunized 1-week-old and adult control mice with rLCMV s.c. and analyzed T cell responses ten days later (Figure 3). For reasons of technical feasibility, the s.c. rather than the i.m. route was used for experiments involving 1-week-old mice. TB10.4-specific CD8 T cell frequencies of 1-week-old mice as determined by MHC class I multimers and intracellular cytokine assays were comparable, if not superior, to those of adult animals (Figure 3A,B). Similarly, 1-week-old mice mounted adult-like Ag85B-specific CD4 T cell responses (Figure 3C), altogether indicating that rLCMV was immunogenic even in early life.

### 2.4. rLCMV Can Be Co-Administered with Human Infant Vaccines

The immunogenicity of vectored *Mtb* vaccination can be impaired when human infant vaccines are co-administered [33]. Hence, we tested a potential interference of rLCMV with a hexavalent Expanded Program of Immunization (EPI) vaccine (DTPa-6), which targets diphtheria, tetanus and pertussis, Hepatitis B, polio and Haemophilus influenzae (Hib) in mice. One-week-old animals were immunized with rLCMV alone, with DTPa-6 alone or simultaneously with both vaccines. Thirteen days later, animals receiving both rLCMV and DTPa-6 exhibited approximately two-fold lower frequencies of TB10.4-multimer-binding CD8 T cells than animals vaccinated with rLCMV alone, and a similar trend was noted when assessing TB10.4-specific CD8 T cells by intracellular cytokine assays (Figure 4A). In contrast, rLCMV-induced CD4 T cell responses to TB10.4 and Ag85B seemed unaffected by DTPa-6 co-administration (Figure 4A). DTPa-6-induced antibody responses to tetanus toxoid (TT) as well as to the toxoid (PT), pertactin (PRN), and filamentous hemagglutinin (FHA) of Bordetella pertussis were not diminished by rLCMV co-administration (Figure 4B). In a complementary approach, we tested whether rLCMV could be co-administered with the DTPa-6 booster. For this, 1-week-old mice were given DTPa-6 and, at eight weeks of age, were boosted with either DTPa-6 plus rLCMV or with DTPa-6 alone. Control groups were not vaccinated at one week of age and at eight weeks of age were given either rLCMV plus DTPa-6 or rLCMV alone (see chart in Figure 4C). DTPa-6 co-administration with rLCMV to 8-week-old mice augmented TB10.4-specific CD8 T cell responses ~2-fold, irrespective of prior DTPa-6 primary immunization. In contrast, CD4 T cell responses to rLCMV-vectored Ag85B were ~2-fold higher in animals receiving only rLCMV at 8 weeks of age compared with mice given DTPa-6 prime in early life and rLCMV plus DTPa-6 when eight weeks old. A similar trend was noted for animals without early life DTPa-6 vaccination and subsequent co-administration of rLCMV and DTPa-6 at 8 weeks of age, suggesting the interference of DTPa-6 with CD4 T cell induction by rLCMV at 8 weeks of age was primarily due to the contemporaneous DTPa-6 administration rather than to early life DTPa-6 priming. Importantly, however, rLCMV co-administration at 8 weeks of age did not interfere with antibody responses to DTPa-6 prime-boost vaccination (Figure 4D). Taken together, these findings indicate that rLCMV co-administered with DTPa-6 priming or boosting does not interfere with antibody induction to the DTPa-6 vaccine. Conversely, DTPa-6 co-administration can modulate CD8 and CD4 T cell responses to rLCMV by up to twofold, either positively or negatively, which may differ between T cell subsets and depend on the age of the vaccinated individual. Co-administered DTPa-6 vaccination does not, however, abrogate rLCMV immunogenicity.

### 2.5. T Cell Responses upon rLCMV- and/or BCG Immunization and Subsequent Mtb Aerosol Challenge

As BCG is currently the only TB vaccine in clinical use, we compared immune responses of mice vaccinated with BCG, homologous rLCMV prime-boost, or heterologous vaccination with BCG followed by two doses of rLCMV (BCG + LCMV). Controls were given phosphate-buffered saline (PBS) instead of the vaccine, and the efficacy of these different vaccine regimens against *Mtb* aerosol challenge was tested in mice (Figure 5A). Two weeks after the last group of animals had completed their respective course of vaccination (Figure 5B, Week 2), the levels of TB10.4-multimer-binding CD8 T cells and Ag85B-tetramer-binding CD4 T cells in the blood of rLCMV-immunized mice exceeded those of BCG-only vaccinated and PBS-treated control animals. BCG itself expresses both Ag85B and TB10.4. Accordingly, the responses observed in BCG + LCMV immunized mice were significantly higher than those after immunization with rLCMV only, indicating that rLCMV effectively boosted BCG-primed T cells. Two to three weeks after aerosol challenge of the animals the aforementioned differences in Ag85B-specific CD4 T cell responses had receded, but TB10.4-specific CD8 T cell responses of mice immunized with rLCMV alone or with BCG + rLCMV remained significantly higher than those of BCG-only-immunized mice and of PBS controls (Figure 5B, Week 2–3). Four weeks after *Mtb* challenge, the latter rLCMV-induced differences in CD8 T cell responses persisted in the spleen (Figure 5B, Week 4). Ag85B-specific CD4 T cell frequencies, however, were similar in rLCMV-immunized and PBS control-treated mice, but animals with BCG or BCG + rLCMV immunization exhibited significantly lower responses than PBS controls, suggesting prior BCG exposure accounted for a shift in antigen/epitope dominance of T cell responses after *Mtb* challenge. In the lung, both TB10.4-specific CD8 T cell and Ag85B-specific CD4 T cell frequencies were the lowest in BCG-vaccinated animals, whereas Ag85B-specific CD4 T cell frequencies in PBS-control vaccinated mice were highest, suggesting the differences observed at this late time point were mostly reflective of vaccination-induced immunodominance hierarchies and, potentially, bacterial loads effects.

### 2.6. rLCMV Immunization Reduces Mtb Loads in Lung, Improves Lung Pathology, and Augments CD8 T Cell Recruitment to the Lung

Next, we determined how the various vaccination regimens impacted *Mtb* loads and resulting histopathological alterations. Immunization with rLCMV or BCG significantly reduced bacterial burdens in the lungs four weeks after *Mtb* challenge (Figure 6A). Combining both vaccines (BCG + LCMV) did not further enhance protection. To determine whether improved *Mtb* control in vaccinated animals translated into ameliorated lung pathology, we relied on computer-assisted evaluation of lung sections, discriminating ventilated from non-ventilated areas (Figure 6B). The percentage of ventilated lung area was significantly lower in PBS control-treated mice than in either rLCMV, BCG, or BCG + LCMV immunized animals (Figure 6C), suggesting all vaccination regiments helped preserve lung ventilation. When performing immunohistochemistry to detect T cell infiltrates at week 4 after *Mtb* challenge, we noticed that CD8 T cells were clearly more abundant in rLCMV-immunized than in PBS control-vaccinated mice (Figure 6D), both in ventilated as well as in non-ventilated regions of the lung. Computer-assisted image analysis revealed that, on average, CD8 T cells were ~2–2.5-fold more numerous in the lungs of either BCG- or rLCMV-immunized animals than in PBS controls (Figure 6E). An analogous trend was noted for animals immunized with BCG+rLCMV. However, CD8 infiltration in the latter mice was significantly lower than in ventilated and non-ventilated lung regions of BCG-only and rLCMV-only-immunized animals, respectively. This may indicate some yet undefined reciprocal interference of the two vaccines. Alternatively, but not mutually exclusively, isolated analysis at week 4 may have failed to reveal differential kinetics of T cell infiltration and *Mtb* clearance, which could also have contributed to such differences between vaccine groups. CD4 T cell infiltrates were also analyzed and showed higher densities in PBS control-treated animals than in either one of the vaccinated groups, again, likely a consequence of the animals’ higher *Mtb* burden (Figure 6F).

## 3. Discussion

The present work indicates that replication-deficient LCMV vectors represent a valuable addition to a growing quiver of vaccination technologies and vaccine candidates, which in combination may eventually afford the long-sought life-long protection against pulmonary tuberculosis. Excellent immunogenicity even in the neonatal period and only limited interference with EPI vaccines suggest that rLCMV may lend itself to inclusion in early life vaccination programs. Later in life, the ability to effectively readminister this vaccine vector without detectable interference by vector-neutralizing antibodies [24] may facilitate the daunting task of maintaining protective anti-*Mtb* immunity. 

Despite expressing only two of the many mycobacterial antigens present in BCG, the rLCMV vaccine offered a level of protection that was comparable with this current gold-standard vaccine, suggesting that the induction of strong T cell responses by means of a viral vector can represent an appropriate strategy for limiting *Mtb* replication. The *Mtb*-specific T cell responses elicited by rLCMV comprised not only strong CD8 but also robust CD4 T cell responses, both of which showed a high degree of functionality as judged based on cytokine secretion patterns. Accordingly, it remains unknown at present whether the protective efficacy of rLCMV vaccination against *Mtb* challenge relied on CD4 or CD8 T cell memory or on a combination of both. 

Upon challenge, CD8 T cells were recruited to the lungs of rLCMV- and/or BCG-vaccinated animals in higher numbers than in non-vaccinated controls, correlating with protection against *Mtb* replication. In contrast, CD4 T cell infiltration density followed an inverse pattern and correlated better with bacterial loads, which were higher in non-vaccinated controls. These observations are in line with previous studies indicating that BCG vaccination induces more rapid accumulation of CD4 T cells in the lung following *Mtb* challenge, whereas at later timepoints, T cell numbers correlate poorly with other measures of protection [34]. The current analyses conducted at a single time point after *Mtb* challenge (4 weeks) may need to be supplemented by studies at earlier timepoints to identify vaccine-induced immune correlates that are decisive for the subsequent course of infection. 

While live mycobacterial vaccines such as BCG generate responses against a broad repertoire of antigens, candidate vaccines expressing select antigens found in both BCG and *Mtb*, such as Ag85B and TB10.4, have the potential to increase protection by boosting the corresponding T cell responses. In our experiments, the combination of BCG and rLCMV in a heterologous prime-boost regimen failed to demonstrate a protective benefit over BCG vaccination alone. This finding was somewhat disappointing given that BCG+rLCMV immunization resulted in pre-challenge TB10.4-specific CD8 T cell frequencies that were ~100-fold higher than upon BCG vaccination alone (50% vs. 0.25%), and a ~10-fold difference persisted in blood at week 2–3 and in spleen at week 4 after *Mtb* challenge (see Figure 5B). These differences in blood and spleen translated into fourfold higher TB10.4-specific CD8 T cell frequencies in the lung at week 4, but overall CD8 T cell densities in the lung of BCG+rLCMV-immunized animals were not different from those of BCG-only vaccinated mice (compare Figure 5B vs. Figure 6E). These observations may indicate that the measurement of TB10.4-reactive CD8 T cells was not a fair reflection of the totality of *Mtb*-specific CD8 T cell immunity elicited by BCG, a response that is expected to target epitopes across the entire bacterial proteome. The immunodominant TB10.4 peptide epitope has been suggested to represent an immunological “decoy”, which failed to be recognized on Mtb-infected macrophages [35]. It, therefore, is possible that rLCMV vectors delivering other *Mtb* antigens may offer better anti-bacterial protection than Ag85B-TB10.4 and may show better synergy with BCG vaccination.

Perhaps more surprisingly, pre-challenge Ag85B-specific CD4 T cell responses in the blood of BCG+rLCMV-vaccinated mice were ~10-fold higher than in BCG only-immunized animals, a difference that was virtually annihilated within the first 2-3 weeks after *Mtb* challenge (Figure 5B). One mechanistic explanation for this observation could consist in these cells’ recruitment to the infected lung. Comparable frequencies of lung-infiltrating Ag85B-specific CD4 T cells in BCG+rLCMV- and BCG only-vaccinated mice at week 4 after challenge did not, however, provide support to this hypothesis. Accordingly, clonal competition with other specificities of BCG-primed CD4 T cells that also responded to challenge seems more plausible. 

Taken together, the present work provides a comprehensive characterization of the rLCMV delivery platform in the context of prophylactic *Mtb* vaccination. It demonstrates a balanced and solid induction of both CD4 and CD8 T cell immunity in adult as well as neonatal animals and significant antibacterial protection in the lung without histological evidence of immunopathology. Future work should investigate the inclusion of novel *Mtb* antigens into rLCMV vectors and address how to optimally combine rLCMV with other immunization regimens to further optimize long-term protection against pulmonary tuberculosis in human adults and/or in infants.

## 4. Materials and Methods

### 4.1. Mice

C57BL/6 mice were originally purchased from Charles River and were bred locally.

### 4.2. BCG and Mtb Production, Mtb Challenge, and Bacterial Titer Determination

*Mycobacterium bovis* BCG SSI 1331 (American Type Culture Collection (ATCC) No. 35733) and *M. tuberculosis* H37Rv (ATCC, No. 27294) stock were prepared as described previously [36]. For the challenge experiments, female, age-matched adult littermates were randomly assigned to treatment groups. No significant differences in body weight were observed between the experimental groups. An aerosol challenge of mice with *Mtb* was performed using a Glas-Col inhalation exposure system to deliver a dose of 100 CFU over the course of 1 h. To quantify *Mtb* loads in lungs, the tissue was homogenized in phosphate-buffered saline supplemented with 0.05% Tween 80 (PBST) and protein inhibitors using GentleMACS dissociator M tubes (Miltenyi Biotec, Bergisch Gladbach, Germany). Serial dilutions were performed in PBST and plated onto Middlebrook 7H11 agar and incubated at 37 °C for 3 to 4 weeks, following which colony-forming units were counted.

### 4.3. Viral Vectors and Vaccines

The rLCMV vector expressing Ag85B-TB10.4 (herein referred to as “rLCMV” throughout) was generated, and stocks were grown and titrated following established protocols [20,21]. The Ag85B-TB10.4 fusion protein consisting of amino acids 42-325 of the Ag85B ORF and fused at its *C*-terminus to the 96 amino acid-long TB10.4 ORF was preceded by the human tissue plasminogen activator signal peptide for optimal immunogenicity [37]. The LCMV backbone was derived from the Armstrong Clone 13 variant, with asparagine at position 400 of the nucleoprotein mutated (N400S) to eliminate the immunodominant H-2D^b^-restricted viral epitope NP396-404 [38]. rLCMV was administered intramuscularly (i.m.) at a dose of 2 × 10^6^ plaque forming units unless specified otherwise. BCG was administered to mice subcutaneously at a dose of 10^6^ CFU in 100 µL PBS. For DTPa-6 immunization a total dose of 100 µL Infanrix Hexa (GlaxoSmithKline, Brentford, UK) was administered simultaneously via the i.m. route (25 µL into each leg) and intraperitoneally (50 µL). 

### 4.4. Determination of Antigen-Specific T Cell Responses

Specific CD4 and CD8 T cell responses were determined by intracellular cytokine staining and MHC multimer staining following established procedures [39]. For intracellular cytokine staining, *Mtb*-specific CD4 T cells were restimulated using either a recombinant Ag85B-TB10.4 fusion protein (GenScript Biotech, Leiden, Netherlands) at a final concentration of 2.5 µg/mL or with the immunodominant Ag85B-derived CD4 T cell peptide epitope Ag85B301-320 (THSWEYWGAQLNAMKGDLQS), and CD8 T cells were restimulated using the immunodominant TB10.4 peptide epitope IMYNYPAM (amino acids 4-11). H-2I-A^b^ tetramers (MHC class II) loaded with the Ag85B epitope FQDAYNYYGGHNAVF (amino acid 280-294) were generously provided by the NIH Tetramer Core Facility. H-2K^b^ dextramers (MHC class I) loaded with the immunodominant TB10.4 epitope IMYNYPAM (amino acids 4–11) were from Immudex, Copenhagen, Denmark. Epitope-specific T cell frequencies were determined after gating on B220^–^CD8^+^ lymphocytes, B220^–^CD4^+^ lymphocytes, or B220^–^CD4^+^CD4^+^CD62L^–^ lymphocytes, as indicated in the figures. For the analysis of lung-infiltrating T cells by flow cytometry, the lungs were digested with collagenase (0.7 mg/mL collagenase IV, Sigma-Aldrich, St. Louis, MI, USA, and 0.3 mg/mL collagenase D, Roche, Basel, Switzerland, in RPMI 1640 medium at 37 °C in 5% CO_2_ for 30 min, and single-cell suspensions were prepared by mechanical dissociation through a 70 μm-pore-size nylon mesh using RPMI 1640 medium with 10% fetal calf serum (Gibco, Thermo Fisher, Waltham, MA, USA). 

### 4.5. Histology, Immunohistochemistry, and Quantitative Assessment of T Cell Infiltration and Lung Ventilation

Mouse tissues were fixed in 4% formalin and embedded in paraffin. Sections were stained with hematoxylin/eosin (H/E) or processed for immunohistochemistry as follows: endogenous peroxidases were inactivated by a peroxidase blocking solution (Dako, Jena, Germany; K0672), and before CD8 staining, an additional block with Fab fragment goat-anti-mouse IgG (JacksonImmunoResearch, West Grove, PA, USA; 115-007-003) was performed to reduce unspecific binding. The sections were then incubated with the following primary antibodies: rat anti-mouse CD8 (Invitrogen, Waltham, MA, USA; 14-0808-82) and rabbit anti-mouse CD4 (Cell Signaling Technology, Danvers, MA, USA; 25229). Bound primary antibodies were stained with HRP conjugated secondary antibodies specific for rat (Vector Laboratories, Burlingame, CA, USA; MP-7444-15) or rabbit (Dako, Jena, Germany; K4003). Bound secondary antibody was revealed with 3,3′-diaminobenzidine as chromogen (Dako, Jena, Germany; K3468) and counterstained with Hemalum (Merck, Darmstadt, Germany; 1.09249.0500) for brightfield microscopy. Slides were scanned using a Pannoramic Flash Scanner (3D Histech, Budapest, Hungary) at 200× magnification. To evaluate the proportion of ventilated and unventilated lung surface, an automated analysis of non-ventilated area was performed on H/E-stained lung sections. For each animal, one representative image of a whole lung section was captured with a 20× objective. Thereafter, an analysis was performed by automatic processing of the images in a custom-programmed script of Cognition Network Language based on the Definiens Cognition Network Technology platform (Definiens Developer XD software Version 2.7; Definiens, Munich, Germany). In brief, the programmed script discriminated lung tissue and tissue-free surroundings by spectral difference detection. The surface of the resulting region of interest (non-ventilated lung area, ROI) was calculated and subtracted from the total lung surface in order to evaluate ventilated lung area for each mouse. 

To determine CD8 and CD4 T cell infiltration densities in lung tissue, the slides were scanned on a Pannoramic 250 Flash II whole slide scanner (3DHistech, Budapest, Hungary) with a resolution of 0.221 µm/px. The images were analyzed using a custom script in Definiens Developer XD 2.7 (Definiens AG, Munich, Germany) consisting of the following steps: The tissue was identified based on pixel intensities. Ventilated and non-ventilated areas of lung were differentiated using morphological open and closing operations on the binary tissue mask image. Within each one of these areas, CD4 T cell and CD8 T cells (on separate sections) were detected based on DAB intensity after color deconvolution of the hematoxylin and DAB colors. The total area per compartment and signal area per cell type were quantified. The numbers of CD4 T cells and CD8 T cells per mm^2^ were calculated.

### 4.6. Determination of DTPa-6-Induced Antibody Responses

To determine antibody responses elicited by DTPa-6 (Infanrix Hexa; GlaxoSmithKline, Brentford, UK), we performed enzyme-linked immunosorbent assays, as previously described [40]. In brief, tetanus toxoid (TT), *Bordetella pertussis* toxoid (PT), *Bordetella pertussis* pertactic (PRN), and filamentous hemagglutinin (FHA) were coated to plates and overlaid with serially diluted mouse serum, and bound antibodies were detected using a mouse IgG-specific detection antibody, the binding of which was quantified by means of an electrochemiluminescent label.

### 4.7. Statistical Analysis

We performed statistical analyses using Graph Pad Prism software. For pairwise comparisons two-tailed unpaired Student’s *t*-tests were performed, whereas for the comparisons of three or more groups, one-way ANOVA with Bonferroni’s post-test was performed. Bacterial loads were log-converted to obtain a near-normal distribution for statistical analysis. *p* values < 0.05 were considered statistically significant, *p* < 0.01 was considered highly significant, and *p* > 0.05 was considered not statistically significantly different (n.s.). 

## Figures and Tables

**Figure 1 ijms-23-02700-f001:**
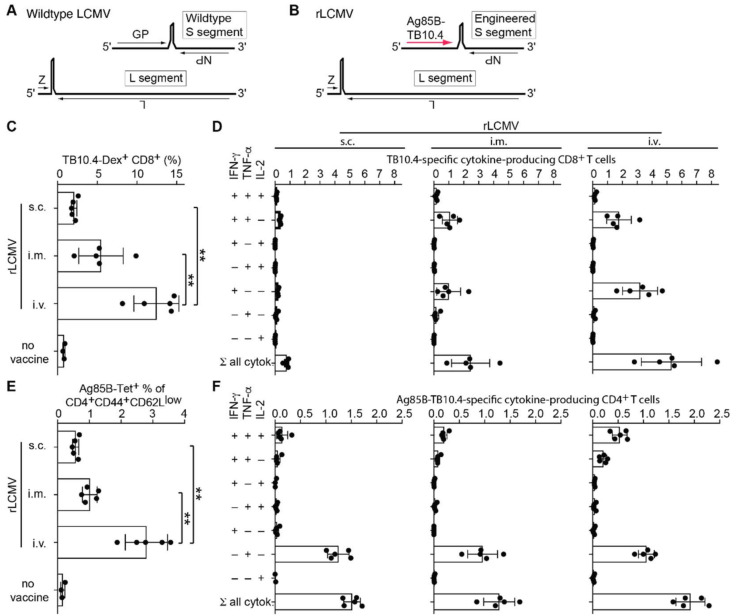
rLCMV elicits high frequencies of polyfunctional CD8 and CD4 T cells. (**A**,**B**): Schematic representation of the wildtype LCMV (**A**) and rLCMV vector (**B**) genomes. (**C**–**F**): We immunized C57BL/6 mice with 10^6^ PFU of rLCMV either s.c., i.m., or i.v. Controls were left unimmunized (no vaccine, panels (**C**,**E**) only). After 28 days, we measured T cell responses to Ag85B-TB10.4 in the spleen. (**C**): TB10.4-dextramer-binding CD8 T cells. (**D**): TB10.4-specific cytokine-producing CD8 T cells. (**E**): Ag85B-tetramer-binding cells amongst activated (CD44^+^CD62L^low^) CD4 T cells. (**F**): Ag85B-TB10.4-specific cytokine-producing CD4 T cells. Bars represent the mean +/− SD of five mice per group. Black circles show individual mice. Values in (**D**,**F**) are background-subtracted (cytokine secretion upon peptide/antigen stimulation minus cytokine secretion in medium-only control wells). One representative experiment of two similar ones is shown. Responses of mice immunized by the i.v., i.m., and s.c. routes were compared by one-way ANOVA with Bonferroni’s post-test. Only statistically significant differences (*p* > 0.05) are indicated. ** *p* < 0.01.

**Figure 2 ijms-23-02700-f002:**
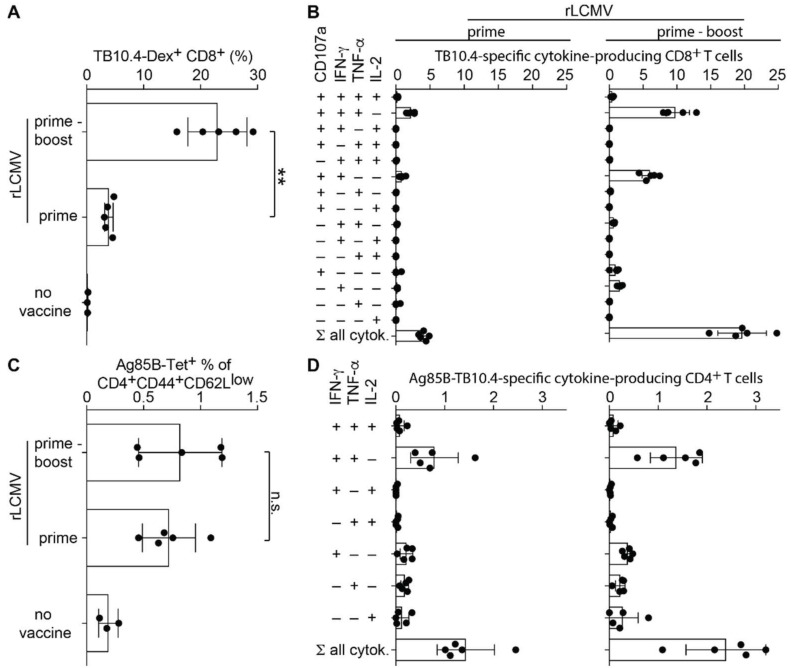
rLCMV-induced CD8 T cell responses are augmented by homologous boosting. We immunized C57BL/6 mice with 10^6^ PFU of rLCMV i.m. on day 0 and day 28 (prime-boost) or on day 28 only (prime). Controls were left unimmunized (no vaccine, panels (**A**,**C**) only). On day 56, we measured T cell responses to Ag85B-TB10.4 in the spleen. (**A**): TB10.4-dextramer-binding CD8 T cells. (**B**): TB10.4-specific cytokine-producing CD8 T cells. (**C**): Ag85B-tetramer-binding cells amongst activated (CD44^+^CD62L^low^) CD4 T cells. (**D**): Ag85B-TB10.4-specific cytokine-producing CD4 T cells. Bars represent the mean +/− SD of five mice per group. Black circles show individual mice. Values in (**B**,**D**) are background-subtracted (cytokine secretion upon peptide/antigen stimulation minus cytokine secretion in medium-only control wells). One representative experiment of two similar ones is shown. Responses of mice immunized with either one or two doses of rLCMV were compared by unpaired two-tailed Student’s *t*-test. ** *p* < 0.01; n.s.: not statistically significant, *p* ≥ 0.05.

**Figure 3 ijms-23-02700-f003:**
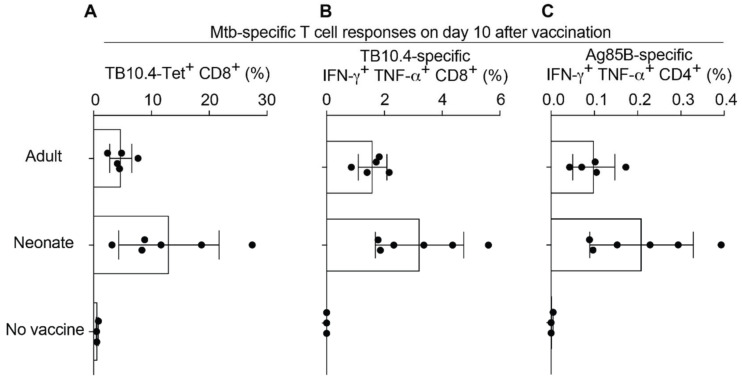
Neonatal mice mount robust CD8 and CD4 T cell responses to rLCMV vaccination. We immunized 1-week-old C57BL/6 mice s.c. with 10^5^ PFU of rLCMV and analyzed T cell responses in spleen 10 days later. Adult C57BL/6 mice immunized s.c. served as positive controls, and unvaccinated adult animals served as negative control (no vaccine). (**A**): TB10.4-dextramer-binding CD8 T cells. (**B**): TB10.4-specific IFN-γ+ TNF- α+ double-producing CD8 T cells. (**C**): Ag85B_301-320_-specific IFN-γ+ TNF-α+ double-producing CD4 T cells. Black circles show individual mice, of which bars represent the mean +/− SD. Values for cytokine-producing T cells are background-subtracted (cytokine secretion upon peptide/antigen stimulation minus cytokine secretion in medium-only control wells). One representative experiment of two is shown.

**Figure 4 ijms-23-02700-f004:**
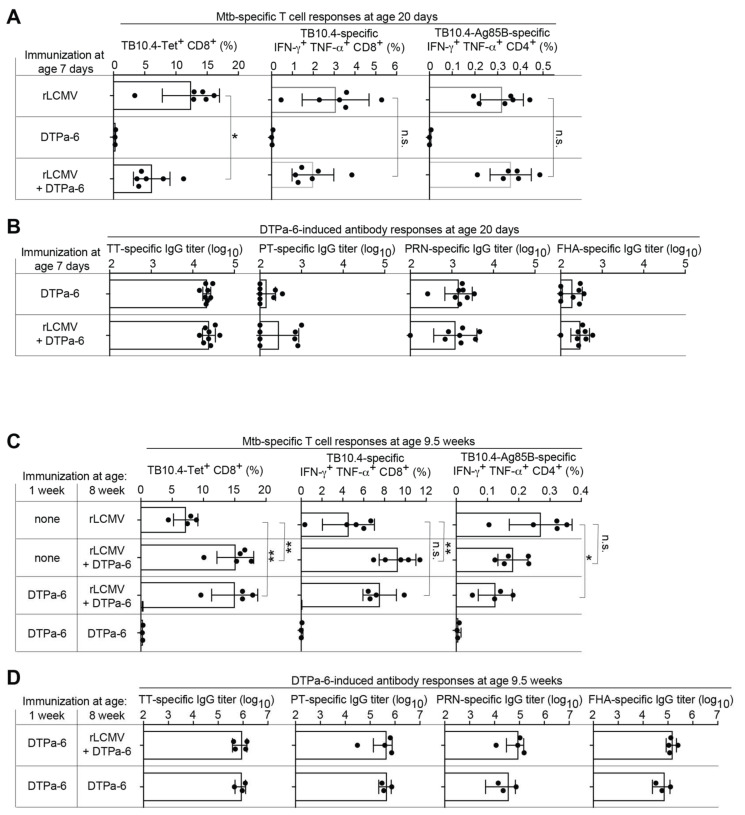
rLCMV can be co-administered with human infant vaccines. A,B: We immunized 1-week-old C57BL/6 mice with 10^5^ PFU rLCMV s.c. (“rLCMV”), with DTPa-6 i.p. and i.m. (“DTPa-6”), or simultaneously with rLCMV plus DTPa-6 by the same respective routes (“rLCMV + DTPa-6”). (**A**): At the age of 20 days, we determined CD8 and CD4 T cell responses in spleen. Left: TB10.4-dextramer-binding CD8 T cells. Center: TB10.4-specific IFN-γ+ TNF-α+ double-producing CD8 T cells. Right. Ag85B-TB10.4-specific IFN-γ+ TNF-α+ double-producing CD4 T cells. (**B**). DTPa-6-induced antibody responses against tetanus toxoid (TT), pertussis toxoid (PT), (**B**): pertussis pertactin (PRN) and pertussis filamentous hemagglutinin (FHA) at the age of 20 days. (**C**,**D**): We immunized 1-week-old C57BL/6 mice with DTPa-6 i.p. and i.m. or left them unvaccinated. At 8 weeks of age, the animals were immunized with rLCMV s.c., with DTPa-6 i.p. + i.m. or were given both vaccines as outlined in the chart. At the age of 9.5 weeks we determined CD8 and CD4 T cell responses in spleen. Left: TB10.4-dextramer-binding CD8 T cells. Center: TB10.4-specific IFN-γ+ TNF-α+ double-producing CD8 T cells. Right. Ag85B-TB10.4-specific IFN-γ+ TNF-α+ double-producing CD4 T cells. (**D**): DTPa-6-induced antibody responses in the groups having received neonatal DTPa-6. Black circles show individual mice, of which bars represent the mean +/− SD. Values for cytokine-producing T cells are background-subtracted (cytokine secretion upon peptide/antigen stimulation minus cytokine secretion in medium-only control wells). One representative experiment of two similar ones is shown. The indicated pairwise comparisons of T cell responses were conducted by unpaired two-tailed Student’s *t*-tests. * *p* < 0.05; ** *p* < 0.01; n.s.: not statistically significant, *p* ≥ 0.05.

**Figure 5 ijms-23-02700-f005:**
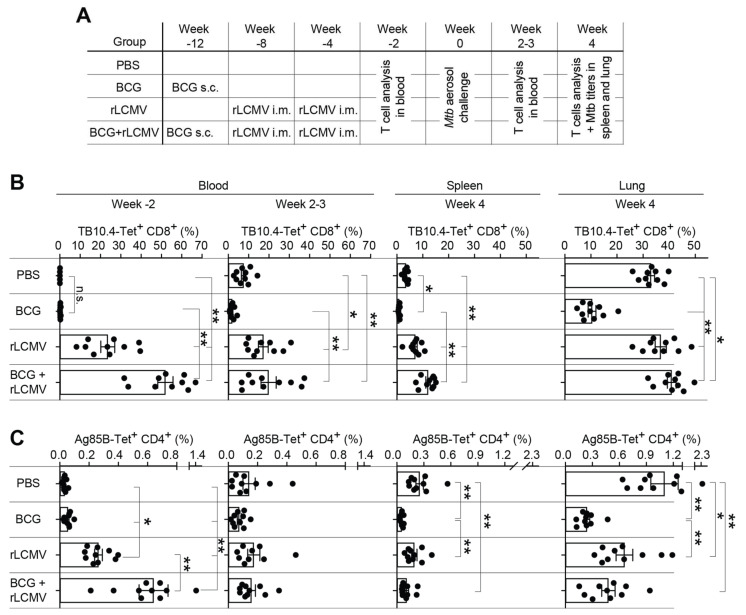
T cell responses upon rLCMV- and/or BCG immunization and subsequent *Mtb* aerosol challenge. (**A**): We immunized adult C57BL/6 mice with BCG s.c. and with rLCMV i.m. at the time points in the combinations indicated in the chart. rLCMV-induced TB10.4-specific CD8 T cells as well as Ag85B-specific CD4 T cells were determined in blood, spleen, and lung at the indicated time points prior to and after *Mtb* aerosol challenge. (**B**). We determined the frequencies of TB10.4 dextramer-binding CD8 T cells 2 weeks prior to and 2–3 weeks after *Mtb* challenge in blood and 4 weeks after *Mtb* challenge in spleen and lung. (**C**): We determined the frequencies of Ag85B tetramer-binding CD4 T cells 2 weeks prior to and 2–3 weeks after *Mtb* challenge in blood, and 4 weeks after *Mtb* challenge in spleen and lung. Symbols in (**B**,**C**) show individual animals from two groups of five mice each that were immunized and challenged independently. Bars show the mean+/- SEM. Data were analyzed by one-way ANOVA followed by Bonferroni’s post-test. * *p* < 0.05, ** *p* < 0.01; only statistically significant differences (*p* > 0.05) are indicated.

**Figure 6 ijms-23-02700-f006:**
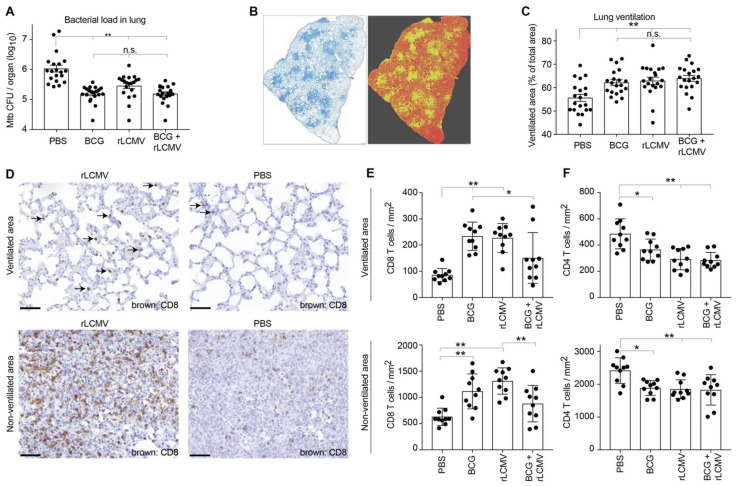
rLCMV immunization reduces *Mtb* loads in lung, improves lung pathology, and augments CD8 T cell recruitment to the lung. Mice were immunized with BCG s.c. and/or with rLCMV i.m. and challenged with *Mtb* as outlined in Figure 5A, and lungs were analyzed at week 4 after challenge. (**A**): Bacterial loads in lung. (**B**): Exemplary microscopy image of a whole-lung cross section stained by hematoxylin/eosin (left). Ventilated and non-ventilated regions were determined by computer-assisted image analysis and are color-coded for illustration purposes in red and yellow, respectively (right). (**C**): The percentage of ventilated surface on whole-lung cross sections, computer-assessed as in (**B**), was calculated as “(ventilated surface):(ventilated + non-ventilated surface)”. (**D**–**F**): Lung sections were processed for immunohistochemical detection of infiltrating CD8 and CD4 T cells. **D**: Representative images from ventilated (top) and non-ventilated (bottom) lung areas of rLCMV-only vaccinated animals and PBS controls. Arrows point out infiltrating CD8 T cells in the alveolar walls of ventilated lung. Magnification bars: 50 µm. (**E**,**F**): Infiltration densities of CD8 (**E**) and CD4 T cells (**F**) in ventilated (top) and non-ventilated (bottom) areas of the lung were determined by computer-assisted image analysis. Symbols in (**A**,**C**) show individual animals from two independently conducted experiments, which each consisted of two groups of five mice that were immunized and challenged independently. Symbols in (**E**,**F**) show results from one of these experiments with two groups of five mice immunized and challenged independently. Bars show the mean+/− SEM. Data were analyzed by one-way ANOVA followed by Bonferroni’s post-test. * *p* < 0.05, ** *p* < 0.01, n.s.: not statistically significant, *p* ≥ 0.05. In (**E**,**F**) only statistically significant differences (*p* > 0.05) are indicated.

## Data Availability

The data presented in this study are displayed in the figures. The raw values can be provided by the authors upon request.

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
