# Peer review of "Replication-Deficient Lymphocytic Choriomeningitis Virus-Vectored Vaccine Candidate for the Induction of T Cell Immunity against Mycobacterium tuberculosis"

_ijms, 2022, doi:10.3390/ijms23052700_

Round 1
Reviewer 1 Report
In the manuscript Replication-deficient lymphocytic choriomeningitis virus-vectored vaccine candidate for the induction of T cell immunity against Mycobacterium tuberculosis" by Elodie Belnoue and colleagues, the authors developed a replication-deficient viral-vector platform based on lymphocytic choriomeningitis virus (rLCMV). The rLCMV vaccine platform was designed to deliver two very-well studied Mtb antigens, Ag85B and TB10.4, and was validated through immunogenicity and efficacy studies in mice.
The authors characterized the immunogenicity in adult and neonatal mice, tested a potential interference of rLCMV with a hexavalent Expanded Program of Immunization vaccine (DTPa-6), by co-administrationg both vaccines, and studied the protective efficacy against an Mtb aerosol challenge.
Even though the new vaccine platform showed an interesting immunogenicity, particularly in the generations of antigen-specific CD8 T cells, and comparable efficacy to BCG, the vaccine failed to improve the efficacy of BCG, when administered as a booster.
The authors claim the rLCMV as an attractive new vector for the delivery of Mtb antigens and as a candidate to be considered for inclusion in neonatal and adult Mtb vaccination regimens.
The study was very well conducted, with the appropriate controls in place, well presented and the discussion covers all the relevant details. The results are very clear, and the conclusions well supported. I have no major concerns about this manuscript.
Minor concerns and comments:
1) Lines 299-301: The authors claim that “Computer-assisted image analysis revealed that, on average, CD8 T cells were ~2–2.5-fold more numerous in the lungs of either BCG- or rLCMV-immunized animals than in PBS controls (Fig. 6E).” Interestingly, in the same panel, the combination of BCG+rLCMV reduced the number of CD8 T cells, to the number comparable to the PBS control. Could this suggest an inhibitory mechanism, caused by the combination of both vaccines, leading to a reduced recruitment of CD8 T cells? Could the authors please elaborate on this question?
2) Could the authors please include a sentence, in line 301, referring to the reduction of recruitment of CD8 T cells upon administration of both vaccines combined?
3) Following on the question 1), given that the immunodominant TB10.4 peptide epitope IMYNYPAM has been described to be a decoy peptide (Samuel Behar´s paper PMID: 29782535), could the authors please elaborate in the Discussion section the possible contribution of this decoy peptide in the inhibition of CD8 T cell recruitment?
4) Lines 125-126 and 150 - Small document formatting typo. Please replace the symbols to the respective Greek letters.
5) Figure 5, the letter “C” is missing from the panel.
Reviewer 2 Report
In this manuscript, Belnoue et al have reported the use of replication deficient choriomeningitis virus (rLCMV) as a vector for the delivery of M. tuberculosis antigens TB10.4 and Ag85B (vaccine). They have also reported that mice immunized with rLCMV delivered vaccine showed enhanced CD8 antigen specific T cells. However, rLCMV delivered TB10.4 and Ag85B vaccine showed efficacy more or less similar to that of TB10.4+Ag85B subunit vaccine developed by Andersen and his colleagues. Unfortunately, rLCMV delivered TB10.4 and Ag85B vaccine in BCG immunized as a booster has no enhanced protection either. Thus, rLCMV may be a new delivery system but with limited scope as a vaccine against TB. Some other comments are below.
- Mice immunized with i.v route show higher immune responses than s.c and i.m routes. But challenge experiments were conducted with mice immunized with s.c route. No rationale for this selection was given.
- Testing of the rLCMV delivered vaccine in neonatal mice and co-administration of the vaccine with infant vaccine seemed to be unnecessary and too early at this stage.
- The figures are weak and dull, and they need to be changed to color.
Reviewer 3 Report
Dear Editor-in-Chief,
I have now read the manuscript for Int.J.Mol.Sci 2022 by Belnoue E et al. entitelled: Replication deficient lymphocytic choriomeningitis virus-vectored vaccine candidate …..
The manuscript presents a significant amount of results and data regarding the efficacy of the rLCMV vaccine vector with and without the BCG vaccine alone or as prime-boost immunization designed M.tb vaccine studies Very young mice (1 week old) were vaccinated and compared with adult female mice. Results indicate a partially protective role of the BCG vaccine, but also the prime booster vaccine BCG+rLCMV vectored vaccine against aerosol-challenge with M.tb.
Questions and comments:
Q1a. The authors have in their Materials and Methods section page 12, in the first paragraph described their aerosol M.tb challenge system as Glas-Col inhalation exposure of challenge dose 100 CFU. How long-lasting was the aerosol-exposure time ? Could the authors explain what the challenge dose of 100 CFU means for young small mice and adult mice used in their study?
Q1b) Could the challenge dose of 100 CFU be to high to allow determination of protective immune responses induced by the vaccines tested?
Q1b) Was there a bodyweight or gender difference between the aerosol challenged mice regarding to how they resisted the Mtb challenge (in Figures 6A and 6C).
Q1d) In Figure 6A in the BCG, rLCMV and in the BCG+rLCMV groups some individual animals had especially low Mtb CFU / organ levels. Did these individuals show a particular immune response that was especially strong in the CD8+ or CD4+ T-cell responses?
